# Advancing crop classification in smallholder agriculture: A multifaceted approach combining frequency-domain image co-registration, transformer-based parcel segmentation, and Bi-LSTM for crop classification

**Waleed Khan** [1,2], **Nasru Minallah** [1,2], **Madiha Sher** [1,2], **Mahmood Ali khan** [1], **Atiq ur Rehman** [3] *, **Tareq Al-Ansari** [4], **Amine Bermak** [3]

**1** National Center for Big Data and Cloud Computing, University of Engineering and Technology, Peshawar, Pakistan, **2** Department of Computer Systems Engineering, University of Engineering and Technology, Peshawar, Pakistan, **3** Division of Information and Computing Technology, College of Science and Engineering, Hamad Bin Khalifa University, Doha, Qatar, **4** Division of Sustainable Development, College of Science and Engineering, Hamad Bin Khalifa University, Doha, Qatar

☯ These authors contributed equally to this work.
* atirehman@hbku.edu.qa

## Abstract

Agricultural Remote Sensing has the potential to enhance agricultural monitoring in smallholder economies to mitigate losses. However, its widespread adoption faces challenges, such as diminishing farm sizes, lack of reliable data-sets and high cost related to commercial satellite imagery. This research focuses on opportunities, practices and novel approaches for effective utilization of remote sensing in agriculture applications for smallholder economies. The work entails insights from experiments using datasets representative of major crops during different growing seasons. We propose an optimized solution for addressing challenges associated with remote sensing-based crop mapping in smallholder agriculture farms. Open source tools and data are used for inter and intra-sensor image registration, with a root mean square error of 0.3 or less. We also propose and emphasize on the use of delineated vegetation parcels through Segment Anything Model for Geospatial (SAM-GEOs). Furthermore a Bidirectional-Long Short-Term Memory-based (Bi-LSTM) deep learning model is developed and trained for crop classification, achieving results with accuracy of more than 94% and 96% for validation sets of two data sets collected in the field, during 2 growing seasons.

## Introduction

Agriculture is the cornerstone of sustainability of a country. In fact, it has enabled the human species to establish settlements and communities, providing the foundation for civilization's

**Data Availability Statement:** Data are available at: https://github.com/khanwaleed1011/Training-Data-CSVs-for-PLOSOne.

**Funding:** This research work is funded in-part by Qatar National Research Fund (QNRF) through grant no. MME01-0922-190049. The findings herein reflect the work and are solely the responsibility of the authors. The funders had no role in study design, data collection and analysis, decision to publish, or preparation of the manuscript.

**Competing interests:** The authors have declared that no competing interests exist.

growth and development. An agricultural state is comprised of a diverse range of geographic landscapes, spanning from fertile plains to deserts, and a diligent populace [1]. The economy of such a state relies heavily on agriculture, serving as the primary source of employment. With the advent of modern-day technology, it has become essential for administrative bodies to monitor agricultural farms using satellite remote sensing, given its ease of access and vast geographical coverage. Satellite-based remote sensing is the observation of planetary and explanatory resources with a deeper insight. Since the inception of the idea of remote sensing, by Ms Evelyn Pruitt of the U.S. Office of Naval Research [2], scientists have taken keen interest to bring advancement in this technology from time to time, by developing application-specific integrated camera sensors for aerial and satellite platforms. These data receivers are embedded in special satellites, placed in a planet's orbit providing coverage of targets or entire planets from time to time. Today remote sensing satellites can provide data with a much better spatial, temporal, radiometric and spectral resolution compared to their predecessors, enabling us to monitor our resources of interest with a focused insight [3]. Remote sensing technologies have emerged as a powerful tool for enhancing agricultural production and sustainability [4] and is pivotal for developing economies with smallholder farms. But implementation of a cost effective and reliable large scale remote sensing system for agricultural monitoring in smallholder economies, is engulfed by many challenges. Accessibility and accuracy of data, small fields and climate conditions are some of major concerns. Nevertheless, the opportunities are vast, and adaptation of this technology can result in substantial enhancement in agricultural productivity and curbing economic losses. To address these issues, remote sensing in conjunction with artificial intelligence can prove to be a valuable tool for enhancing accuracy monitoring crops while addressing the problem of smallholder farms over vast spatial areas [5].

## Remote sensing and deep learning

Remote Sensing data in conjunction with deep learning techniques has created windows of opportunity for research and development in this field, thus resulting in development of statistical systems for the agriculture sector. These satellites offer a range of features such as different channel numbers with specific wavelengths, as well as spatial, temporal, and spectral resolutions, and revisit times. While the high costs associated with specialized satellite imagery can be a barrier, a remarkable number of satellites offer free and accessible data, making them invaluable assets for remote sensing. Noteworthy among these are the Sentinel programs of the European Space Agency (ESA) and NASA's missions like MODIS and Landsat. offering freely accessible data, these satellites enable a diverse range of applications in agriculture monitoring, and providing cost-effective and widespread remote sensing-based solutions [6]. The past decade has witnessed the rise of Deep learning as a powerful tool used in diverse range of areas, including remote sensing. [7]. Inspired by the learning capabilities of the human brain, Artificial Neural Networks (ANNs) are powerful tools that have been used to solve many different problems [8]. This work analyse major challenges setting back implementation of a reliable remote sensing system for classification of crops in smallholder economies, crucial for acreage estimations.

LSTM is one of the most used techniques in Recurrent Neural Networks (RNNs) for sequential or time series data. Therefore, LSTM allows the evaluation of plantations' phenological variation, detecting pixel coherence between time sequence data. Among RNNs, the LSTM model is widely used to capture time correlation efficiently [9]. Bidirectional-LSTM is a modified version of LSTM, which learns from the data from both directions instead of one, thus making it more susceptible to crop growth parameters. Bi-LSTM models are more effective as their output depends on the previous and the next segment, in contrast to the

unidirectional LSTM models. Many studies have been carried out to improve the LSTM architecture. One of the most powerful approaches is Bi-LSTM [10]. Bi-LSTM models are generally more effective when handling contextual information since their output at a given time depends on both previous and next. Bi-LSTM is selected for its outstanding ability to effectively capture and model temporal dependencies within sequential data from both sides, aligning seamlessly with the dynamic characteristics of agricultural landscapes. While some of the other popular supervised machine learning approaches include Random Forest (RF) [11]; Support Vector Machines (SVM) [12] and K-Nearest Neighbor (KNN) [13]. RF learns data through binary trees, with low number of parameters, but with a huge variation in results. SVM is by nature binary classifier but is also used for multi class classification. The approach categorizes using linear or nonlinear functions and has several parameters to be tuned. KNN is a classic and widely used algorithm for remote sensing. This approach does not include the training phase and is based on majority vote mechanism for the provided data, using Euclidean distance. The decision to employ the Bi-LSTM method in this study is based on a deliberate assessment of the unique features present in agricultural remote sensing data.

## Challanges for smallholder agriculture

1. Lack of Reliable datasets: Using state of the art artificial intelligence algorithms such as deep learning requires a large amount of ground truth data for training. Such datasets need to be created, curated, and refined, which is rather unavailable or inaccessible.

2. Geometric error in multi sensor satellite images, poses a greater challenge for accurate classification and acreage estimation and are often erroneous due to spatial mixing between the fields.

3. Small holdings: Due to the growth in population and land fragmentation, land holdings are diminishing in size [1, 14]. This problem makes the utilization of open data satellites challenging as statistics generated are erroneous.

4. Spatial Resolution: Each pixel covering the per unit area on the ground i.e., Spatial Resolution of Open data satellites is not adequate compared to the field sizes.

5. Cloud Cover Challenge: Varying climatic conditions in different regions e.g. cloud cover. These clouds can hinder the capturing of satellite images, leading to difficulties in acquiring reliable and consistent data.

Numerous studies have been performed to tackle challenges in smallholder farms, and a key focus has been on prioritizing Unmanned Aerial Vehicles (UAVs) equipped with multispectral sensors [15, 16]. However, these approaches face fundamental limitations, notably in terms of scalability and economic constraints. Basic limitations for such approaches, are geographic scalability and economic constraints. In light of the aforementioned challenges, this research endeavors to address them systematically, offering solutions at each step and presenting optimized outcomes from experiments conducted on the collected datasets. Mobile and web-based application (GeoSurvey) is utilized for collecting and curating these data sets over a vast geographical area encompassing smallholder farms. Geo-Survey is indiginoulsy developed and is free to use with its companion web-based application. Synergy of two remote sensing satellite systems is created utilizing high resolution Sentinel-2 and very high resolution Planet Scope Dove. Sentinel-2 images are re-sampled to Planet Scope using Nearest Neighbour approach to retain the original information in Sentinel-2 data channels. For multi-sensor and co-registration phase correlation approach is used, which is a well-established method for

image co-registration [17]. Pre-trained Vision Transformer (ViT) encoder and a lightweight transformer-based decoder for segmentation of land parcels is utilized [18]. The manuscript also propose the adoption of short temporal frames (STF) from the phonological cycle of major crops, hence minimizing the possibility of cloud cover.

Additionally, the study includes a comparative analysis between Bi-LSTM and LSTM, with the goal of assessing their individual performance. This analysis aims to provide insights into the effectiveness of these recurrent neural network architectures for the specific task under investigation.

The rest of the manuscript entails:

Related Work of the manuscript delves into the existing body of research in the field, providing an insightful comparison of various studies and their findings followed by area of experimentation. The approach to data collection and analysis is explained in Material and Methods, revealing the specifics of data sources and collection methodologies, following which Bi-Directional Long Short-Term Memory (Bi-LSTM) model for crop classification is described. Next is the section outlining the process through which the segmentation of vegetation parcels was carried out, a crucial aspect of the research. Finally, the manuscript presents the outcomes and findings of the study, allowing readers to grasp the significance of the research within the broader context of the field. These sections collectively contribute to a comprehensive understanding of the study's methodology and results.

## Related works

Remote sensing satellites are paving the way for sustainable agricultural practices by facilitating optimized resource allocation, yield prediction, and crop loss mitigation [19, 20]. A meticulously engineered satellite network, comprising GeoEye-1, Ikonos, Formosat-2, SPOT 6 and 7, and ALOS, empowers decision makers with detailed data for informed agricultural decisions [21, 22]. Table 1 highlights the most frequently used remote sensing satellites for vegetation monitoring.

The progress made in machine learning (ML) and computer vision has led to the creation of advanced models that utilize remote sensing data for tasks like crop mapping, classification, and yield prediction. As a result, farmers can gain valuable insights into their crops, including their spatial distribution, specific types, and even predictions of future yields. Remote sensing

**Table 1. Frequently used remote sensing satellites for vegetation monitoring.**

| Open Data Satellites | | | | |
|---|---|---|---|---|
| | Multispectral Channels | Highest Spatial Resolution in Meters | Temporal Revisit Time | Scene Width in km |
| Sentinel-2 | 13 | 10m | 5days | 60 km |
| Landsat 8-9 | 11 | 15m | 16days | 185 km |
| Aster | 14 | 15m | 16days | 60 km |
| Modis | 36 | 250m | Twice Daily | 2330 km |
| Commercial Satellites | | | | |
| Planet-Scope | 8 | 3m | 1 day | 24.6 x 16.4 |
| World View3 | 16 | 1.24m | 4.5 days | 13.1 km |
| Skysat | 5 | 1m | 6-7 daily | 1 x 2.5 sq km |
| Ikonos | 5 | 1m | 3-5 days | 15 km |
| Geo Eye | 4 | 0.41m | 2-3 days | 20km//14 km |
| Quick Bird | 5 | 0.61m | 1-5 days | 13 km |
| Gaofen 1-2 | 5 | 0.81m | 5 days | 11 km |
| Rapid Eye | 5 | 5m | 5.5 days | 77 km |

**Table 2. A comparative analysis of state of the art.**

| Name | Dataset(s) | Proposed Methodology | Pilot Region | Achieved Results |
|---|---|---|---|---|
| Ji S et al. | GF2 2015 GF2 2016 | Pixel-based classification Comparison of 2D CNN, 3D CNN with SVM, KNN and PCA+KNN | Guyuan City | 3D CNN with Overall Accuracy of 93.9% for GF2-2015 and 95.59% for GF2-2016 |
| Zhang et al | Indigenous | Pixel-based Classification U-Net compared with Customized UNET | Guyuan City | Customized U-Net achieved and Accuracy of 97% |
| Filho et al | Indigenous | Pixel Based Classification Comparison of Bi-LSTM with LSTM and other machine learning models. | Brazil | An accuracy of 99% is achieved for Bi-LSTM compared to 98% of LTSM |
| Lozano-Tello et al. | Indigenous | Pixel Based Classification Comparison of Sentinel-2 12 band with NDVI of the respective imageries. | Iberian Peninsula | 93.34% accuracy is achieved for 12 band Sentinel-2 while 91.41% for the derived NDVI. |
| Gao et al. | Indigenous | A2SegNet Mapping of Corn and Soyabean | United States | Overall validation accuracy of 91.53% was achieved with respect to 92.68% for 2020 and 94.19% for 2021 |
| Wang et al. | Indigenous | Bi-LSTM10 Land Cover Classes | China | Test Accuracy of 85% |
| Mohanrajan et al. | Indigenous | Novel Vision Transformer Based Bi-LSTM | India | Test Accuracy of 98% |

technologies hold the potential to transform the approach to agricultural production, revolutionizing management practices and leading to advancements in food security while simultaneously decreasing the environmental footprint of farming. A comparative analysis of stat of the art is provided in Table 2.

Recurrent Neural Neural Networks (RNN)s are a type of ANNs used for dealing with sequential or time series data. Among the types of RNN models, Long Short Term Memory (LSTM) has been found to be significantly effective in handling time series data problems [23, 24]. LSTMs are designed to tackle long-term dependency problems that occur between events with time gaps, and they excel at learning and remembering information [25]. They do not face optimization challenges, which are a significant issue for Simple RNNs [26]. LSTM models are well suited to solving problems like handwriting recognition, speech synthesis, analyzing audio and video data and language translation [27–29].

To characterize the different periods of crop growth, Ji S et al. idealizes 3D Convolutional Neural Networks (3D-CNN) for 2 data sets used in their study: GF2-2015 and GF2-2016. A new technique involves using spatio-temporal remote sensing images combined with 3D-CNNs instead of 2D-CNNs for crop mapping [30]. U-net has been widely used in image classification tasks. Zhang et al. proposes an enhanced version of U-net architecture and use it for semantic segmentation through their collected data. A synergy of a multispectral single date and Synthetic Aperture Radar (SAR) data is used for classification [31]. SAR-based analysis and classification is performed using machine learning models for paddy detection by Filho et al. LSTM and Bi-LSTM is used for comparison purpose. The results reveal Bi-LSTM to be superior performer in terms of accuracy with respect to other employed machine learning models and LSTM [10]. A sentinel-2 based time series data is compared with their derived Normalized Difference Vegetation Index (NDVI), for crop classification in the region of Iberian Peninsula, Spain. Lozano Tello et al performed surveys in their region of interest and 10 major crop classes were collected as training data for Neural Network models [32]. A study on soyabean and corn cultivation is performed in the regions of Great Lakes of United States. Attention mechanism A2SegNet was trained on a time series dataset of 3 years Sentinel-2 data. The model was trained separately for 2019, 2020 and 2021 datasets, with varying accuracies, i.e., for 2019 an overall validation accuracy of 91.53% was achieved with respect to 92.68% for 2020 and 94.19% for 2021 [31, 33]. A comprehensive analysis of Bi-LSTM for Land Cover Land Use (LCLU) in China was carried out by Wand et al. in regions of China spanning from 1982-2015 [34]. The overall testing accuracy shows a figure of 85% for 10 land cover classes,

for such long time series data, using Bi-LSTM. Mohanrajan et al proposes the use of novel vision transforms-based BI-LSTM for land cover classification in Javadi Hills, India. The overall accuracy achieved using their approach gained an accuracy of 98% [35].

Additionally, Palchowdhuri et al. [36] elaborated on the use of the Random Forest (RF) Classifier [11] for crop type classification using multi-temporal images of WorldView [37] and Sentinel-2, resulting in an overall accuracy of 91%. In their work, C. Pelletier et al. [38] demonstrated the effectiveness of Temp-CNNs for crop type mapping through multi-temporal satellite data, achieving an overall accuracy of 93.5% and predicting 13 different classes. A pixel-based crop classification model was created using TempCNNs and RNNS, considering spectral and temporal features, temporal inception blocks, and fusion of multi-satellite imagery to produce more abundant and richer features, resulting in an accuracy of 98% [39]. M. Weiss et al. presented an empirical methodology to estimate plant traits and agronomical variables using remote sensing in the agricultural sector [40]. They also provided a comprehensive overview of the latest remote sensing techniques specifically designed for agricultural monitoring. Several specialized satellites, such as GeoEye-1, FORMOSAT-2, ALOS, QuickBird, Ikonos, Geo-Fen, SPOT 6, SPOT 7, and World View, are employed to monitor agriculture-related activities such as drought estimation, vegetation detection, crop health assessment and crop classification [21, 41, 42]. Deep CNNs have shown improved performance in crop classification compared to traditional ML methods [43]. Using satellite time series data, a recent study assessed the efficacy of a Temporal Deep Convolutional Model (TempCNN) [38]. Performance comparison of TempCNN with traditional ML algorithms was performed. These are RF, temporal convolutional neural networks (TempCNN) approach and RNN, which is specifically designed for temporal data. The study's findings revealed that TempCNN exhibited better classification accuracy for satellite time series data when compared to the others. A novel dataset, CropDeep, was introduced in [38] for Deep Learning-based studies of diverse vegetation. The dataset is comprised of images captured by diverse equipment such as cameras and IOT devices. The study applied various Deep Learning models and compared their performance. The conclusion drawn was that while Deep Learning algorithms have the potential to achieve significant performance in classifying different crops, there is room for improvement in the algorithms.

Ijabs et al. examined Latvia's region for crop classification utilizing Sentinel-1 and Sentinel-2. The work focuses on the importance of Common Agriculture Policy (COP). They assessed both regular LSTM and Bi-LSTM models, achieving an accuracy of 89.1% for Bi-LSTM, as opposed to 88.34% for LSTM. This reflects a slight improvement of 1.1% observed during the testing phase [9]. Portalés-Julià et al. also emphasize the importance of remote sensing in effectively implementing Conservation of Plots (COP) for sustainable agriculture [44]. They utilized Sentinel-2 time series data to identify abandoned land parcels in their pilot region of Valencia, a community in Spain. Three scenarios were examined, employing eight classification techniques for performance comparison. The techniques included Linear Discriminant Analysis (LDA), Quadratic Discriminant Analysis (QDA), KNN, Decision Trees, Random Forest (RF), LSTM, and BI-LSTM. Across all three scenarios, BI-LSTM demonstrated superior performance with accuracies of 96.1%, 91.8%, and 98.2%. The pixel-based approach was employed for training and testing data splitting. While focusing paddy detection using Synthetic Aperture Radar (SAR) data of Sentinel-1, Filho et al compares traditional machine learning algorithms such as SVM, RF, KNN and Normal Bayes with LSTM and Bi-LSTM [10]. A total of 28000 pixels were used as samples and a train and test split of 19600 and 4400 pixels was used for classification. The results showed the highest accuracy of 99.14% for Bi-LSTM, 98.39% for LSTM, 98.39% for RF, 98.28% for SVM, 97.70% for KNN and 97.46% for NB classifier.

While significant efforts have been devoted to crop mapping, there has been insufficient emphasis on the preprocessing techniques essential for smallholder agriculture. Our proposed methodology highlights the utilization of open-source libraries and tools for addressing key challenges in working with smallholder farms. Our novel approaches includes; In situ collection over a vast geographical region for 2 crop seasons, the use of phase correlation for image registration of multi sensor data to ensure spatial correlation between corresponding pixels in time series images, pre trained vision transformers for parcel extraction to compensate the medium resolution Sentinel-2 for smallholder farms through vegetation parcels and employment of Bi-LSTM based deep learning model for crop classification. Our proposed work builds on these previous studies by using a combination of Planet-Scope and Sentinel-2 data to achieve a higher classification accuracy. Additionally, our proposed model is specifically designed to handle the challenges of smallholder agriculture with precision.

## Experimental area

Area chosen for experiments, covers 4 major districts of Khyber Pakhtunkhwa, Pakistan namely, Swabi, Mardan, Peshawar, Nowshera and Charsadda. The agricultural production of Pakistan, contribute 20-22% of Gross Domestic Product (GDP) to the economy of the country, with a production area of 22 million hectares (ha) [45]. With a population of over 230 million, it is becoming difficult to sustain its own agricultural production, giving an opportunity for imports of variety of agricultural commodities [46]. Being a developing country with a surging population, governments are struggling to cope with multiple problems in this sector. Some of these include over hording by stakeholders, growth of counterfeit crops and unlicensed or unaccounted growth of cash crops like tobacco.

The lack of state-of-the-art technological machinery in the country is one of the primordial reasons for these problems. The persistence of outdated farming methods is a major cause of low yields and financial struggles for smallholder [1], making the region a pivotal point for our experimentation. Surveys were conducted in a vast geographical area covering a total area of 580000 hectares. Our developed surveying application "Geo Survey" was used for the collection of ground truth data. The application is freely available on both Android and IOS. 5 surveys were planned for Kharif season (*DatasetA*) while 4 surveys were conducted during Rabi season (*DatasetB*) in these four districts over a span of one year. Fig 1 shows the observational area as well as the localities of In situ collection. The collective training and testing data is provided in Table 3.

## Experimental setup

### Sentinel-2 and planet-scope

The launch of ESA's Sentinel-2A and 2B, twin Remote Sensing satellites made a significant progress in the availability of open data for Remote Sensing applications, providing Multispectral Images (MSI) consisting of 13 spectral bands. The spatial resolution of these images ranges from 10 to 60 meters per pixel. These satellites are equipped with a range of channels, including 10-meter resolution RGB and NIR channels, 20-meter resolution Vegetation Red Edge 1-4 and SWIR 1-2 channels, and 60-meter resolution Coastal Aerosol, Cirrus, and Water Vapor channels. This diverse set of channels enables the capture of detailed information across different wavelengths, supporting applications such as land cover classification, vegetation monitoring, and atmospheric analysis. The revisit cycle of Sentinel-2 satellites is 5 days, providing frequent and consistent monitoring.

Planet-Scope is small sat-based commercial constellation of MSI satellites, delivering very high resolution (VHR) imagery. These satellites capture MSI with exceptional detail. The VHR

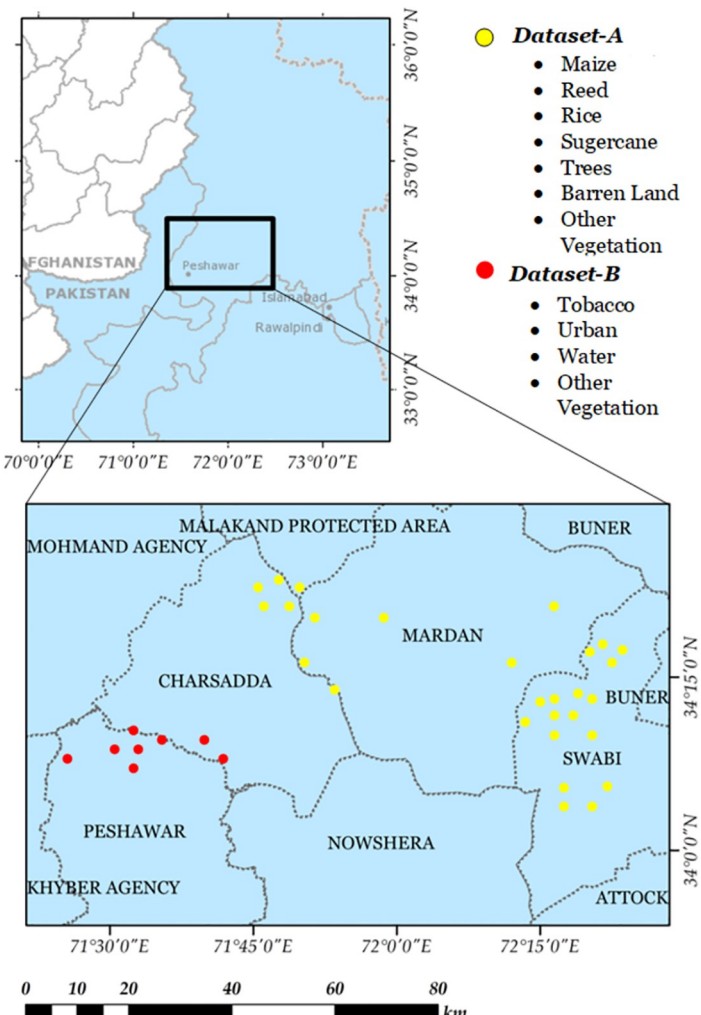

**Fig 1. Area of experimentation in Khyber Pakhtunkhwa (KP), with Red markers highlighting data collection points for *DatasetA* while the Blue marker identifies localities for *DatasetB*.**

**Table 3. Collected ground truth data statistics for *DatasetA* and *DatasetB* seasons.**

| Season | Category | Training Pixels | Testing Pixels |
|---|---|---|---|
| *DatasetA* | Maize | 14000 | 3271 |
| | Reed | 6832 | 1962 |
| | Rice | 7714 | 1925 |
| | Sugarcane | 38227 | 17182 |
| | Trees | 20774 | 6778 |
| | Barren Land | 1606 | 750 |
| | Other Vegetation | 2658 | 806 |
| *DatasetB* | Tobacco | 69,884 | 30264 |
| | Urban Settlements | 50793 | 23276 |
| | Water | 19808 | 4311 |
| | Other Vegetation | 139,571 | 45550 |

**Table 4. Tiles of Sentinel-2 and Planet-Scope Dove used in the study.**

| Tile IDs of Sentinel-2 | Tile IDs of Planet-Scope |
|---|---|
| S2B_MSIL2A_20220508T055629_N0400_R091_T43SBT_20220508T081118 | 20210914_055702_37_240a |
| S2B_MSIL2A_20220508T055629_N0400_R091_T43SBU_20220508T081118 | 20210914_055704_72_240a |
| S2B_MSIL2A_20220508T055629_N0400_R091_T42SYC_20220508T081118 | 20220427_052113_01_2231 |
| S2B_MSIL2A_20220508T055629_N0400_R091_T42SYD_20220508T081118 | 20220427_052110_80_2231 |
| S2A_MSIL2A_20220513T055641_N0400_R091_T43SBT_20220513T111914 | 20220427_052108_59_2231 |
| S2A_MSIL2A_20220513T055641_N0400_R091_T43SBU_20220513T111914 | 20220427_051457_66_2276 |
| S2A_MSIL2A_20220513T055641_N0400_R091_T42SYC_20220513T111914 | 20220427_051455_38_2276 |
| S2A_MSIL2A_20220513T055641_N0400_R091_T42SYD_20220513T111914 | 20220427_051453_10_2276 |
| S2B_MSIL2A_20220518T055639_N0400_R091_T43SBT_20220518T082531 | 20220427_050244_91_2223 |
| S2B_MSIL2A_20220518T055639_N0400_R091_T43SBU_20220518T082531 | 20220427_050242_84_2223 |
| S2B_MSIL2A_20220518T055639_N0400_R091_T42SYC_20220518T082531 | 20220427_050240_77_2223 |
| S2B_MSIL2A_20220518T055639_N0400_R091_T42SYD_20220518T082531 | |
| S2B_MSIL2A_20220528T055639_N0400_R091_T43SBT_20220528T083104 | |
| S2B_MSIL2A_20220528T055639_N0400_R091_T43SBU_20220528T083104 | |
| S2B_MSIL2A_20220528T055639_N0400_R091_T42SYC_20220528T083104 | |
| S2B_MSIL2A_20220528T055639_N0400_R091_T42SYD_20220528T083104 | |
| S2A_MSIL2A_20210816T055641_N0500_R091_T42SYC_20230213T164031 | |
| S2A_MSIL2A_20210816T055641_N9999_R091_T42SYD_20230416T154847 | |
| S2B_MSIL2A_20210821T055639_N0500_R091_T42SYD_20230120T070541 | |
| S2B_MSIL2A_20210821T055639_N0500_R091_T42SYC_20230120T070541 | |
| S2A_MSIL2A_20210826T055641_N0500_R091_T42SYD_20230218T042136 | |
| S2A_MSIL2A_20210826T055641_N0500_R091_T42SYC_20230218T042136 | |
| S2A_MSIL2A_20210925T055641_N0500_R091_T42SYC_20230119T094854 | |
| S2A_MSIL2A_20210925T055641_N9999_R091_T42SYD_20230419T202843 | |

images acquired from Planet-Scope satellites find application in diverse fields such as urban planning, precision agriculture, and natural resource management. Planet's primary mission is to make up-to-date imagery accessible to a wide user base, supporting a range of applications [47]. This commercial satellite constellation offers four core bands: Red, Green, Blue, and Near-Infrared (NIR), delivering VHR with a spatial resolution ranging from 3 to 5 meters. In terms of temporal resolution, Planet-Scope boasts a remarkable capability, capturing images daily, resulting in a temporal resolution of 24 hours. The Sentinel-2 and Planet-Scope Dove data retrieved from https://dataspace.copernicus.eu and www.Planet.com is provided in Table 4.

## Methods

Experimental were performed on Precision 7820 Tower, powered by Intel Xeon Silver 4208, 64 Gigabyte of RAM and Nvidia Quadro RTX 8000 with a 48GB of memory. Fig 2 highlights the major operations and the workflow of our proposed methodology.

The collected data is in the form of polygons with a field identifier as meta data, later used for model training of our deep learning model. Post processing of data is done in a web-based portal of the application. Pre-processing tasks, such as identification and removal of redundant data, purging of conflicting polygons and converging is performed using the GeoSurvey-Web portal. Fig 3 shows a snapshot of mobile version of the application and GeoSurvey-Web portal. After processing the next stage of the process is splitting of the pre-processed ground truth

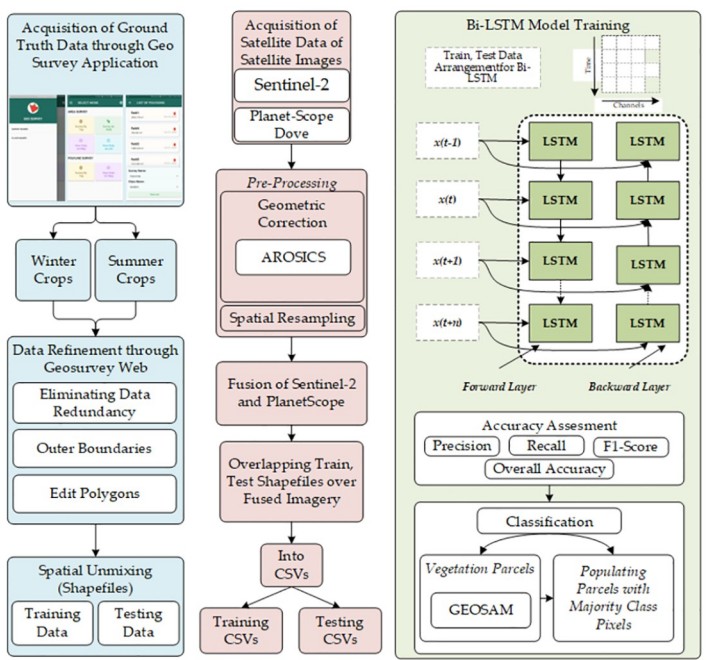

**Fig 2. Step-wise operations of the proposed methodology.**

data into training and testing for deep learning algorithm. Although common or classical practices found in artificial intelligence-based remote sensing literature, is based on the pixel-based splitting of data into training and testing. This approach raises a model accuracy integrity question, due to spatial mixing of training and testing data pixels within the same field, i.e., same pixels of a vegetation field can be found in both training and testing data. To avoid this, we have developed and included a polygon level splitting technique in GeoSurvey-web. This way, we could split the pre-processed ground truth data polygon wise into training and testing sets based on their percentage.

CSVs are generated by projecting the pre-processed training and testing polygons over the created time series satellite data for *DatasetA* and *DatasetB* separately. These CSV are re-arranged using numpy arrays for Bi-LSTM input layer.

Time series of Sentinel-2 and Planet-Scope Dove are acquired from their respective portals, ESA Copernicus datahub and Planet.com. Due to the varying climatic conditions of Pakistan, it is pertinent to note that cloud cover can be a major hurdle for the multispectral remote

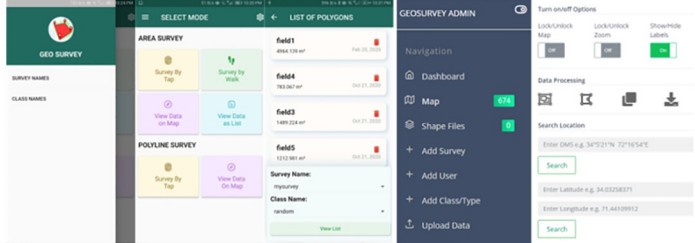

**Fig 3. Representing GeoSurvey application and GeoSurvey-web portal for pre-processing of collected field data.**

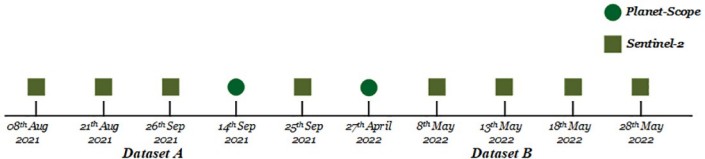

**Fig 4. Short temporal frames for the phonological cycle of *DatasetA* and *DatasetB* Crops.**

sensing. Keeping in view the heterogeneous climatic conditions, short temporal data from these satellites is required for the matured months of *DatasetA* and *DatasetB* crops. This approach is robust, efficient, and implementable on a large geographical area. Sentinel-2 and Planet-Scope fusion is created for this purpose. Data is re-sampled using Nearest Neighbor approach to keep data integrity and avoid the use of interpolated data. A timeline of the acquired Sentinel-2 and Planet-Scope data is provided in Fig 4.

Since 30/03/2021 a geometric refinement step is used to improve the multi-temporal geolocation performance of Sentinel-2. In a first step, the geometric refinement has been applied over Europe and Africa. Since 23/08/2021, the geometric refinement is applied globally using GRI (Global Reference Image) by ESA. Preliminary validation results indicate the following performances for the refined products.

- Absolute geo location: better than 6 m multi-temporal co-registration same repeat orbit: better than 5 m at 95

- Multi-temporal co-registration, different repeat orbits: better than 5 m.

For vegetation studies this number needs to be minimized, for smallholder farms. AROSICS (Automated and Robust Open-Source Image Co-Registration Software) is a significant advancement in geospatial data processing. It employs an approach based on image matching in the frequency domain, along with a multistage workflow (Fig 5), to efficiently recognize and eliminate false positives. For image co-registration, integration of GDAL and OpenCV libraries are employed. AROSICS demonstrates remarkable robustness, making it well suited for large datasets and challenging distortion scenarios. The reference and input images undergo frequency domain conversion using Fourier transform, enabling the identification of spatial shifts. The frequency domain image spectra undergo cross-correlation, with the peak of the cross-correlation function indicating the relative offset between the images The geometric correction using AROSICS is used for both Sentinel-2 and Planet-Scope images.

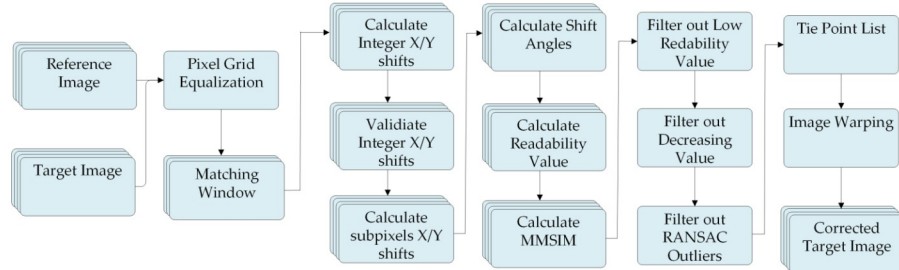

**Fig 5. Automated and Robust Open-Source Image Co-Registration Software (AROSIC) processing chain for geometric correction.**

## Bi-LSTM

Bidirectional Long Short-Term Memory (Bi-LSTM) is a variant of Long Short-Term Memory (LSTM), a RNN. RNNs are used for sequential problems, like natural language processing, speech recognition or time series data. Bi-LSTM is a powerful tool in neural networks, widely used for handling sequences of data. What sets it apart is its ability to understand context from both the past and the future within a sequence. This is done by combining two important things: the LSTM, which remembers information, and looking at data in both directions. Combining LSTM's memory and bidirectional approach, makes it great for understanding sequences of data. LSTM, an advanced type of RNN, addresses the vanishing gradient problem through its dedicated memory cells equipped with gates. These gates control the flow of information, enabling LSTM to retain and utilize data over long sequences. This feature is crucial for interpreting the underlying context and relationships within sequential data. While traditional LSTMs move through sequences in a one-way manner, Bi-LSTMs embark on a dual path. They handle data in two directions, going from the beginning to the end (called the forward pass) and from the end to the beginning (referred to as the back-ward pass). This two-sided method equips the network to capture relationships in both directions, resulting in a more thorough understanding of the context for each element in a sequence. A schematic of the LSTM in provided in Fig 6. The key mathematical components of an LSTM include:

$$i^{(t)} = \sigma(W_i \cdot [h(t-1), x(t)] + b_i) \tag{1}$$

$$f^{(t)} = \sigma(W_f \cdot [h(t-1), x(t)] + b_f) \tag{2}$$

$$o^{(t)} = \sigma(W_o \cdot [h(t-1), x(t)] + b_o) \tag{3}$$

$i^{(t)}$ is the Input Gate, while $f^{(t)}$ is formulates the Forget Gate. $x(t)$ represents the input at time step $t$. $h(t-1)$ is the hidden state from the previous time step. $\sigma$ represents the sigmoid function, and $W_i$, $W_f$, $W_o$ are weight matrices, while $b_i$, $b_f$, $b_o$ are bias vectors.

$$C(t) = \tanh(W_c \cdot [h(t-1), x(t)] + b_c) \tag{4}$$

$$C(t) = f(t) \cdot C(t-1) + i(t) \cdot C(t) \tag{5}$$

$$h(t) = o(t) \cdot \tanh(C(t)) \tag{6}$$

Where $C(t)$ represents the long memory with a *tanhasactivation* and $h(t)$ is the hidden state.

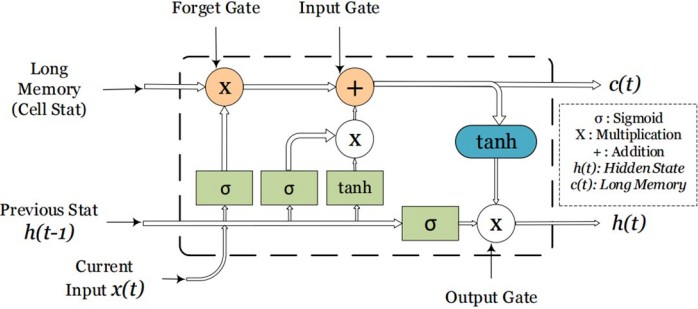

**Fig 6. A representation of long short term memory block.** $x(t)$ represents the input at time step $t$. $h(t-1)$ is the hidden state from the previous time step.

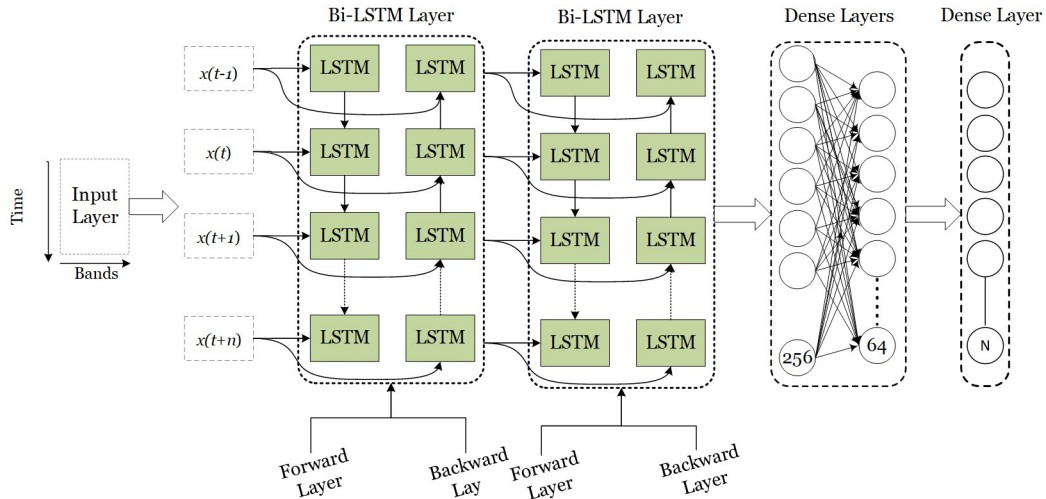

**Fig 7. Our model of Bi-LSTM with two Bi-LSTM, dropout and final classification layers.**

## Model specification

The proposed model starts with an Input Layer, handling data organized in a certain way, like a sequence with five-time steps and four features in each step. The Bidirectional Layer is a key part, using two LSTM layers that can look at data from both the past and the future to understand connections in both directions. This gives an output with a specific shape (Fig 7). Dropout Layers are used to prevent the model from getting too specialized and to keep it flexible. A Flatten Layer simplifies the data, and Dense Layers with different numbers of neurons help make predictions. The neural network configuration includes the following hyper parameter settings: a batch size of 256, 64 filters, a dropout rate of 0.5, tanh as the activation function, Adam as the optimizer, and a learning rate of 0.0001. Together, these parameters shape the architecture and training dynamics of the BI-LSTM, chosen after rigorous tuning of different hyper-parameters (Table 5).

Evaluating model configurations on *DatasetsA* and *DatasetB*, the investigation primarily centers around identifying optimal hyper-parameter combinations based on Training accuracy, Validation Accuracy, with due consideration to Training Loss and Validation Loss. After detailed analysis, it is noted that the activation function "Relu" consistently yields better performance across various scenarios. When employing 2 layers with 64 filters and the Adam optimizer, the Relu activation function attains the highest Training and Validation Accuracy, reaching 97.61% and 96.82%, respectively while also achieving the lowest Training and Validation Loss of 0.0783 and 0.1038, respectively.

The performance across different numbers of filters and layers further reinforces the superiority of the Relu activation function. Thus, based on the outcomes presented in Table 5, it can be inferred that the Relu activation function, in conjunction with 2 layers, 64 filters, and the Adam optimizer, constitutes the optimal hyper-parameter combination, demonstrating superior accuracy and efficiency in both training and validation phases.

While observing the *DatasetB* results, a parallel trend emerges in favor of the Relu activation function. In comparison with *DatasetA*, the 2 layers, 64 filters, and Adam optimizer configuration, coupled with Relu activation, continues to exhibit best performance. The highest Training and Validation Accuracy of 94.76% and 94.62%, is achieved using the same hyper-

**Table 5. Comparison of different parameters for BiLSTM.**

| | | Dataset A | | Dataset B | |
|---|---|---|---|---|---|
| | | Training | Validation | Training | Validation |
| Activation Function (Layers: 2, Filters: 64, Optimizer: Adam) | Sigmoid | 63.08 | 55.44 | 63.76 | 72.12 |
| | Softplus | 91.32 | 88.43 | 93.62 | 94.46 |
| | Softsign | 93.74 | 90.7 | 93.9 | 94.31 |
| | Tanh | 94.8 | 92.9 | 94.08 | 94.53 |
| | Selu | 96.59 | 94.93 | 94.08 | 94.65 |
| | elu | 94.7 | 92.39 | 93.93 | 94.54 |
| | Relu | 97.61 | 96.82 | 94.76 | 94.62 |
| Filters (Activation Function Relu, 2 Layers, 64 filters | 8 | 85.77 | 88.74 | 83.2 | 89.25 |
| | 16 | 93.15 | 92.14 | 92.05 | 94.4 |
| | 32 | 97.36 | 96.51 | 93.47 | 94.64 |
| | 64 | 97.61 | 96.82 | 94.76 | 94.62 |
| | 128 | 95.23 | 92.8 | 94.39 | 94.69 |
| Function (Activation Function: Relu, Layer: 2 Adam, Filters: 64 | SGD | 60.94 | 55.44 | 59.28 | 56.09 |
| | Nadam | 94.21 | 91.41 | 94.15 | 94.31 |
| | Adagrad | 66.09 | 55.44 | 59.17 | 56.09 |
| | Adadelta | 55.61 | 54.26 | 59.16 | 56.09 |
| | Adamax | 91.28 | 89.45 | 93.02 | 94.63 |
| | Adam | 97.61 | 96.82 | 94.76 | 94.62 |
| (Activation Function: Relu, Optimization function: Adam, Filters: 64) | 1 | 93.93 | 92.14 | 94.22 | 94.89 |
| | 2 | 97.61 | 96.82 | 94.76 | 94.62 |
| | 3 | 96.83 | 94.4 | 94.22 | 94.48 |
| | 4 | 94.2 | 92.29 | 94.15 | 94.45 |

parameters, while demonstrating superior performance with the lowest Training and Validation Loss of 0.1692 and 0.1473, respectively. Consequently, mirroring the findings in *DatasetA*, it can be deduced that the Relu activation function, when applied in tandem with 2 layers, 64 filters, and the Adam optimizer, stands out as the optimal hyper-parameter combination for maximizing accuracy and minimizing loss in both *DatasetA* and *DatasetB*. Fig 8 the graphs representing grid search for best hyperparameters based on Training Accuracy, Validation Accuracy, Training Loss and Validation Loss for Bi-LSTM.

## Vegetation parcels delineation

Automatic field delineation is the process of segmenting farm parcels from satellite imagery. It is an important task for precision agriculture, as it can be used to create accurate and up-to-date farm boundaries. In recent years, there has been a growing interest in developing automatic field delineation methods [48]. One of the most promising new methods for automatic field delineation is the SAM model, developed by Meta [18]. Pre-trained Vision Transformer (ViT) encoder and a lightweight transformer-based decoder for segmentation of land parcels is utilized [18]. SAM is a deep learning model that is trained on a large dataset of satellite imagery and farm parcel labels. The model can segment farm parcels with high accuracy, even in challenging conditions, such as when the imagery is noisy or when the fields are irregular in shape. An overview of on our experimentation on vegetation parcel segmentation in area of interest can be seen in Fig 9.

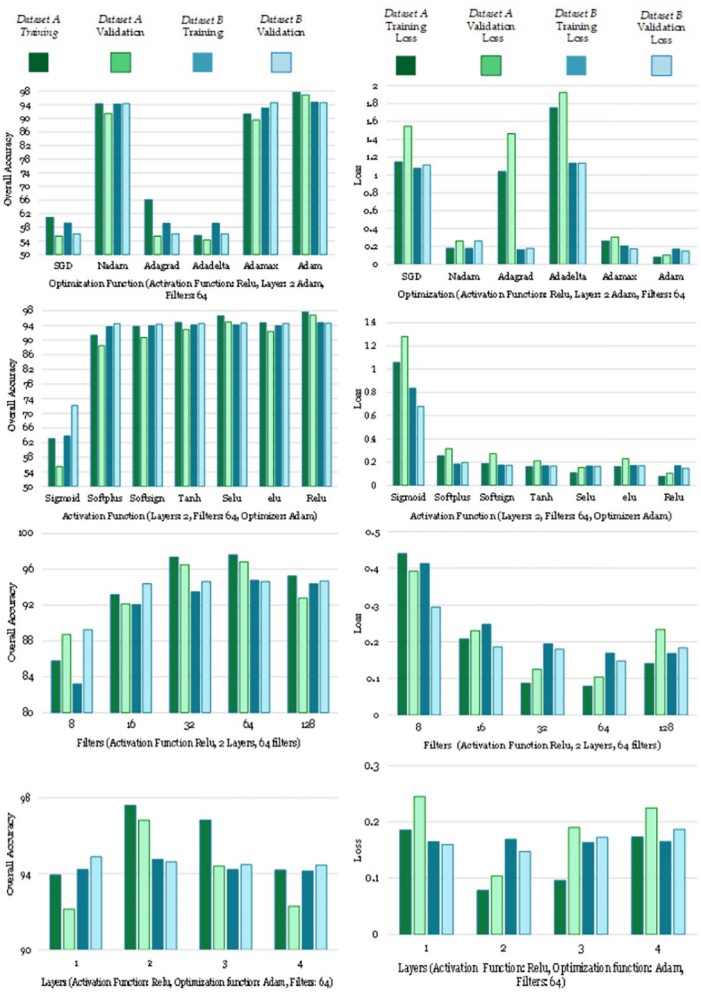

**Fig 8. Grid search for best hyper-parameters setup.**

## Results

2 datasets are created using ground truth data collected during field surveys. Both the datasets have been preprocessed for refinement, split into training and testing field polygons, over-lapped over satellite time series for *DatasetA* and *DatasetB* (Table 3) and finally converted into CSVs, the required format for model training. 2 time series datasets of fused Sentinel-2 and

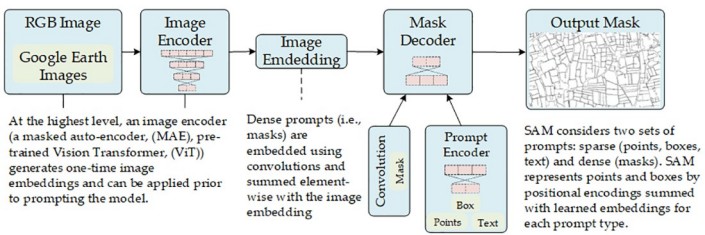

**Fig 9. Vegetation parcel segmentation on SAM model.**

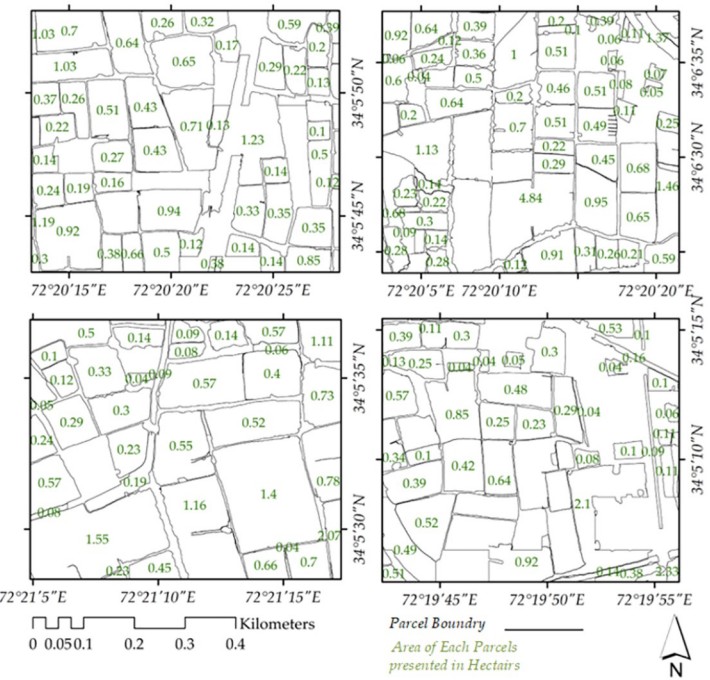

**Fig 10. Segmented parcels in hectares (ha) through SAM.**

Planet-scope has been created, for crop identification Fig 4. The preprocessing steps performed on satellite images include multi-sensor image registration, spatial resampling using nearest neighborhood and time series. A Bi-LSTM-based model is devised and trained for crop classification, also performing its comparison with LSTM. Vegetation parcels delineation is performed using (ViT) encoder and a lightweight transformer-based decoder.

### Image registration

The process of image registration involves geometric correction, specifically between Sentinel-2 scenes and among multi sensors such as Sentinel-2 and Planet Scope.

### Vegetation parcel segmentation

Parcels segmented through Meta's SAM are in the form of polygons, presented in Fig 10, showcasing different locations from the area of experimentation with their extracted vegetation parcels, that are further refined for removal of noisy polygons. As can be seen from the figure the fine boundaries of each polygons provide a detailed representation of the field distribution in the observed area. This segmentation not only allows for a clear visualization of distinct locations but also facilitates the subsequent refinement process, eliminating noisy polygons and ensuring a more accurate depiction of the landscape.

### Validation criteria

The validation criteria used in the study are precision, recall, f1scope and overall accuracy. Table 6 gives the model evaluation criteria used in the study for Bi-LSTM. Precision, Recall, F1-Score, and Overall Accuracy serve as vital metrics applied in diverse do-mains, such as data science and machine learning. Precision gauges the precision of positive predictions, while

**Table 6. Model evaluation criteria for Bi-LSTM.**

| Evaluation Criteria | Representation |
|---|---|
| Precision | $\frac{TP}{TP+FP}$ |
| Recall | $\frac{TP}{TP+FP}$ |
| F1-Score | $2 * \left( \frac{Recall \; * \; Precision}{Recall \; + \; Precision} \right)$ |
| Overall Accuracy | $\frac{\left( Number \; of \; all \; correctly \; classified \; pixel \right)}{\left( Total \; number \; of \; Pixels \right)}$ |

recall evaluates the capacity to capture all pertinent instances. The F1Score harmoniously combines these metrics to strike a balance between precision and recall. Meanwhile, Overall Accuracy quantifies the proportion of correctly classified pixels out of the total pixel count, making it an invaluable gauge for tasks like image classification. These metrics are indispensable for the evaluation of algorithm and model performance, ensuring the trustworthiness of outcomes. A two-step validation is performed in the study;

1. Through split test data at the time of model training

2. By visiting the fields through guided classification maps
   Where *TP* is True Positive and *FP* is False Positive.

## Performance of Bi-LSTM during model training

**Dataset A.** In the training phase, the model starts with an accuracy of 0.745 and steadily improves with each epoch as can be seen in Fig 11(a). This progression is indicative of the model's ability to learn from the training data, with accuracy increasing to 0.976 over the course of training. On the other hand, the validation phase shows how well the model generalizes to unseen data. The validation accuracy begins at 0.832 and, like training ac-curacy, exhibits fluctuations across epochs. It reaches a peak of 0.968, demonstrating that the model performs well on data it hasn't been explicitly trained on. In the training phase, the model's

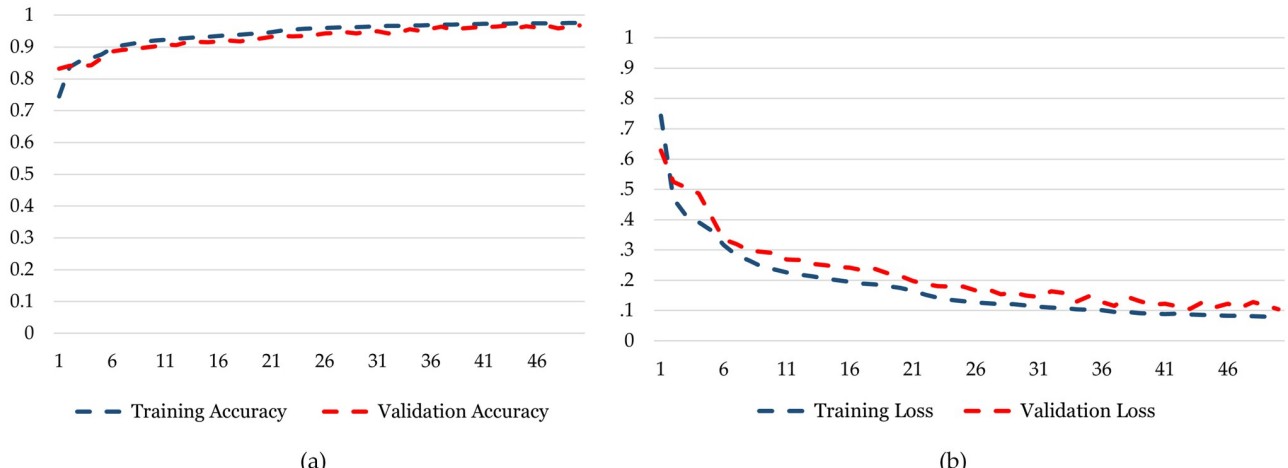

(a)

(b)

**Fig 11. Training and testing graphs for Bi-LSTM.** (a) Training Accuracy and Validation Accuracy for *DatasetA*. (b) Training Loss and Validation Loss for *DatasetA*.

**Table 7. Classification report of Bi-LSTM for *DatasetA* observed under the study.**

| Class | Urban | Other Vegetation | Maize | Reed | Rice | Sugarcane | Trees | Water | Macro Average | Weighted Average | |
|---|---|---|---|---|---|---|---|---|---|---|---|
| | | | | | | Classification Report of Bi-LSTM on *DatasetA* | | | | | |
| Precision | 0.96 | 0.95 | 0.981 | 0.87 | 0.83 | 0.97 | 0.94 | 1 | 0.94 | 0.96 | **Validation Accuracy** |
| Recall | 0.99 | 0.93 | 0.89 | 0.82 | 0.96 | 0.97 | 0.95 | 0.99 | 0.94 | 0.96 | **0.96** |
| F1-score | 0.98 | 0.94 | 0.93 | 0.85 | 0.89 | 0.97 | 0.95 | 0.99 | 0.94 | 0.96 | **Validation Loss** |
| Support | 7282 | 5516 | 3262 | 1917 | 1915 | 13329 | 2788 | 14891 | 50900 | 50900 | **0.1** |

loss steadily decreases as it learns from the data (Fig 11(b). It commences at 0.744 and exhibits a consistent decline, ultimately reaching a low value of 0.078. This signifies that the model is effectively minimizing its training error during the training process. In contrast, the validation phase focuses on the model's performance on unseen data. The validation loss starts at 0.629, showing some fluctuations but generally follows a downward trend. It reaches a minimum value of 0.103, indicating that the model generalizes well to new, unseen data with a minimal validation error.

As can be seen from the classification report of Validation data in Table 7. In the "Urban" class, the model exhibits a precision of 0.96, signifying its ability to accurately predict this category with 96% accuracy. A high recall of 0.99 indicates that the model effectively captures nearly all actual instances within this class, resulting in a well-balanced F1-Score of 0.98. Transitioning to the "Other Vegetation" category, the model maintains a strong precision of 0.95, reflecting 95% accuracy in its predictions. The recall of 0.93 underscores the model's capacity to correctly identify 93% of actual instances, contributing to a solid overall F1-Score of 0.94. The classification report expands its evaluation across diverse classes, delivering detailed insights into the model's performance within each category. The macro average signifies a balanced performance, evidenced by an F1-Score of 0.94, whereas the weighted average addresses class imbalances and showcases even higher F1-Scores of 0.96. Furthermore, the reported validation accuracy of 0.96 demonstrates the model's proficiency in accurately classifying 96% of the validation data. The low validation loss of 0.10 indicates effective model training and precise predictions, contributing to the overall assessment of its performance.

**Dataset B.** The performance of the Bi-LSTM model can be seen in Fig 12(a) and 12(b). When we study the training and validation accuracy patterns in Fig 12, we observe an

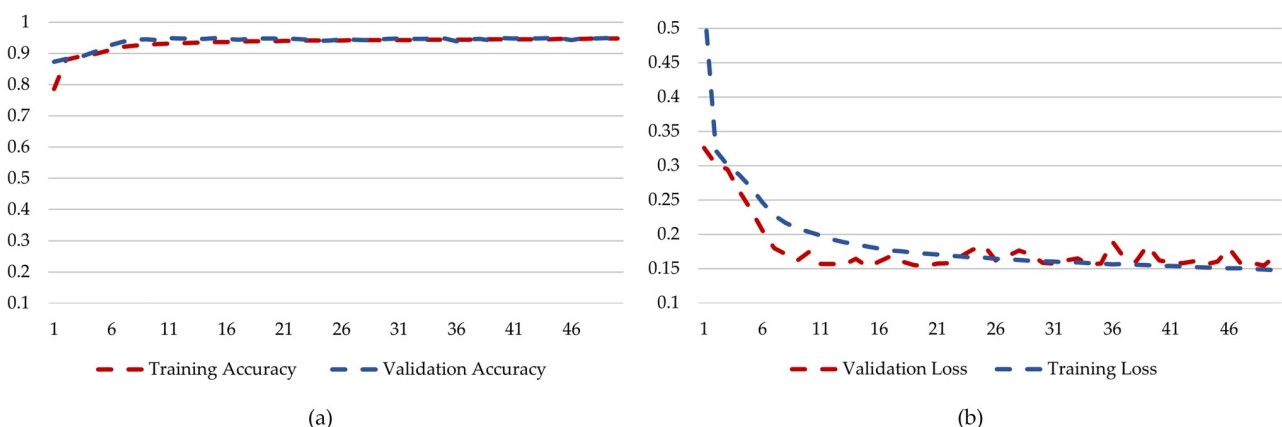

(a)

(b)

**Fig 12. Training and testing graphs for Bi-LSTM.** (a) Training Accuracy and Validation Accuracy for *DatasetB*. (b) Training Loss and Validation Loss for *DatasetB*.

**Table 8. Classification report of Bi-LSTM for *DatasetB* observed under the study.**

| Class | Tobacco | Other Vegetation | Urban | Water | Macro Average | Weighted Average | |
|---|---|---|---|---|---|---|---|
| | | | Classification Report of Bi-LSTM on *DatasetB* | | | | |
| Precision | 0.94 | 0.93 | 0.96 | 1 | 0.96 | 0.94 | **Validation Accuracy** |
| Recall | 0.9 | 0.97 | 0.91 | 0.97 | 0.94 | 0.94 | **0.94** |
| F1-Score | 0.92 | 0.95 | 0.93 | 0.98 | 0.95 | 0.94 | **Validation Loss** |
| Support | 22835 | 59057 | 19425 | 3965 | 105282 | 105282 | **0.16** |

appealing way of how the model improves. At the beginning, the training accuracy is quite low at 0.78, but it steadily gets better with each training round, reaching an impressive 0.94 accuracy. In contrast, the validation accuracy starts at 0.8733, showing potential from the start. As the model encounters new, unseen data during testing, its accuracy consistently improves, reaching an impressive 0.94. This suggests that the model is balanced, performing well on the training data, and making accurate predictions on new data. The most important thing to note is that both training and validation accuracy end up at high values. This indicates that the model effectively learns from the training data and applies its knowledge to new challenges. This shows that the model is skilled and adaptable in various real-world situations, making it a valuable tool for crop classification tasks. In analyzing the training loss and validation loss in, it is evident that the model exhibits distinct trends across multiple epochs. The training loss starts relatively high at 0.54 in the first epoch and then consistently decreases with each subsequent epoch. This decline in training loss suggests that the model is learning and improving its performance over time. On the other hand, the validation loss, which represents the model's performance on unseen data, also starts reasonably high at 0.32 but follows a more erratic pattern. It initially decreases, indicating that the model is generalizing well.

The classification report of Bi-LSTM for testing data provided is in Table 8. In the Tobacco category, the model excels with a notable precision of 0.94, showcasing its ability to make accurate positive predictions, complemented by a 0.90 recall rate. This harmonious blend of precision and recall results in an F1-Score of 0.92. Shifting to the Other Vegetation class, the model impressively achieves a recall rate of 0.97, denoting its accurate identification of instances within this category and culminating in a high F1-Score of 0.95. In the Urban class, the model maintains a commendable balance with a 0.96 precision and a 0.91 recall, reflecting an F1-Score of 0.93. Within the Water category, the model achieves a perfect precision of 1.0, indicating 100% precision, combined with a 0.97 recall, resulting in an exceptional F1-Score of 0.98. The macro average, around 0.95 for precision, recall, and F1-Score, underscores well-rounded performance. The weighted average, considering class imbalances, affirms the model's consistent performance across diverse data distributions. Additionally, a 0.94 validation accuracy highlights the model's ability to classify 94% of the validation data correctly, while a low validation loss of 0.16 showcases its effective training and precision in predictions.

## Performance of LSTM during model training

**Dataset A.** In contrast to the BiLSTM, the LSTM model displays a slightly divergent training pattern. Commencing with a training accuracy of 66.2%, the LSTM model incrementally enhances its performance across epochs, ultimately reaching a 95.1% (Fig 13). Although the LSTM model attains a high accuracy level, it seems to converge at a comparatively slower rate in comparison to the Bi-LSTM. The validation accuracy also demonstrates improvement, initiating at 75.4% and peaking at 94.4%, indicating effective generalization. Both the training and validation loss curves for the LSTM model exhibit a consistent decline, showcasing the model's

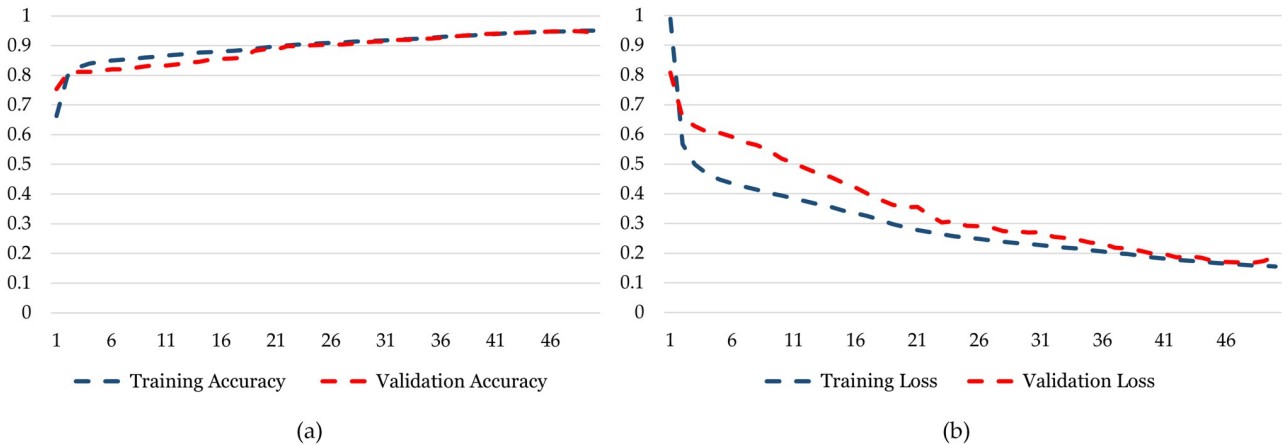

**Fig 13. Training and testing graphs for LSTM.** (a) Training Accuracy and Validation Accuracy for *DatasetA*. (b) Training Loss and Validation Loss for *DatasetA*.

adeptness at error minimization. Comparing BiLSTM and LSTM suggests that BiLSTM can understand more detailed patterns, making it converge faster and potentially be more robust.

The LSTM model for *DatasetA* demonstrates good performance, achieving an overall accuracy of 94.36% (Table 9). It showcases precision, recall, and F1-score metrics for specific classes like 'Urban,' 'Sugarcane,' and 'Water,' attaining precision scores of 98%, 97%, and 100%, respectively. Nevertheless, challenges arise in accurately classifying 'Reed' and 'Rice,' evident in lower precision, recall, and F1-score values compared to Bi-LSTM. While both models proficiently categorize land cover types, the Bi-LSTM model appears to hold a slight advantage, particularly in terms of precision and overall average metrics. The classification of model trained with LSTM on Dataset A is provided in Table 9 below.

**Dataset B.** Unlike the BiLSTM model, the LSTM model has a different way of performing. Although it gets better during training, reaching a highest accuracy of 93.66%, it's still lower than what the Bi-LSTM model achieved as shown in Fig 14. The validation accuracy goes up to 93.75%, following a similar pattern. While the LSTM model is okay in terms of accuracy, the difference between training and validation accuracy suggests it might be getting too good at the training data (overfitting) but looking at the loss numbers, the training loss keeps dropping, but the validation loss stops at 0.1838. This hints that the LSTM model might struggle to apply what it learned to new data as well as the Bi-LSTM model. The LSTM model does well but doesn't quite match the accuracy and adaptability of the Bi-LSTM model for *DatasetB* as well.

As can be seen in Table 10 the LSTM model for *DatasetB* demonstrates commendable performance in the classification report, achieving good precision, recall, and F1-score values

**Table 9. Classification report of LSTM for Dataset B observed under the study.**

| | | | | | | | | | | | |
|---|---|---|---|---|---|---|---|---|---|---|---|
| **Classification Report of LSTM on DatasetB** | | | | | | | | | | | |
| **Class** | **Urban** | **Other Vegetation** | **Maize** | **Reed** | **Rice** | **Sugercane** | **Trees** | **Water** | **Macro average** | **Weighted average** | |
| **Precision** | 0.98 | 0.93 | 1 | 0.63 | 0.62 | 0.97 | 0.89 | 1 | 0.88 | 0.95 | **Validation Accuracy** |
| **Recall** | 1 | 0.92 | 0.85 | 0.57 | 0.87 | 0.95 | 0.93 | 1 | 0.89 | 0.94 | **0.9436** |
| **F1-score** | 0.99 | 0.93 | 0.92 | 0.6 | 0.73 | 0.96 | 0.91 | 1 | 0.88 | 0.94 | **Validation Loss** |
| **Support** | 7282 | 5516 | 3262 | 1917 | 1915 | 13329 | 2788 | 14891 | 50900 | 50900 | **0.1621** |

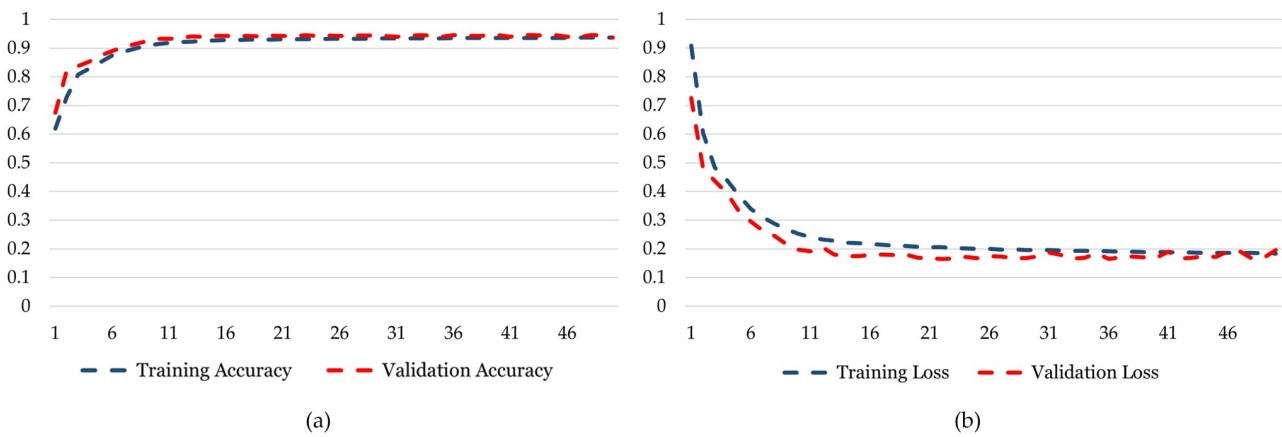

**Fig 14. Training and testing graphs for LSTM.** (a) Training Accuracy and Validation Accuracy for *DatasetB*. (b) Training Loss and Validation Loss for *DatasetB*.

across Tobacco, Other Vegetation, Urban, and Water categories. The precision and recall for the Urban class are somewhat lower compared to Bi-LSTM. Notably, the Water class achieves perfect precision and a respectable recall, indicating the model's competence in identifying water-related features. The LSTM model achieves an overall accuracy of 93.75%, precision of 93.99%, recall of 93.75%, and F1-score of 93.71%. While these metrics slightly trail behind those of the Bi-LSTM model, they exhibit the LSTM model's capability in classifying different features within the dataset.

## Visual results

The classification generated through our trained Bi-LSTM is processed through segmented vegetation parcels, shown in the Fig 15 below. A majority pixel-based filter is applied to all the parcels generated, with the original underlying classification image. A comparison of the final product with the original one provides us with labeled vegetation parcels, using which field acreage estimations can be calculated effectively.

## Conclusion

The implementation of remote sensing-based systems for agricultural production forecasting holds immense potential for agrarian economies with smallholder. To curb the economic loses in smallholder economies, it becomes imperative to leverage state-of-the-art technologies for the accurate generation of crop statistics, through the methodology proposed by this research. This methodology not only aids in informed decision making but also addresses critical issues such as over hoarding, smuggling, and the illicit growth of non-edible crops like tobacco. It

**Table 10. Classification report of LSTM for Dataset B observed under the study.**

| | | | | | | | Classification report of LSTM for Dataset B |
|---|---|---|---|---|---|---|---|
| **Class** | **Tobacco** | **Other Vegetation** | **Urban** | **Water** | **Macro average** | **Weighted average** | |
| **Precision** | 0.93 | 0.93 | 0.96 | 1 | 0.96 | 0.94 | **Validation Accuracy** |
| **Recall** | 0.92 | 0.97 | 0.87 | 0.97 | 0.93 | 0.94 | **0.9375** |
| **F1-score** | 0.92 | 0.95 | 0.92 | 0.99 | 0.94 | 0.94 | **Validation Loss** |
| **Support** | 22835 | 59057 | 19425 | 3965 | 105282 | 105282 | **0.196** |

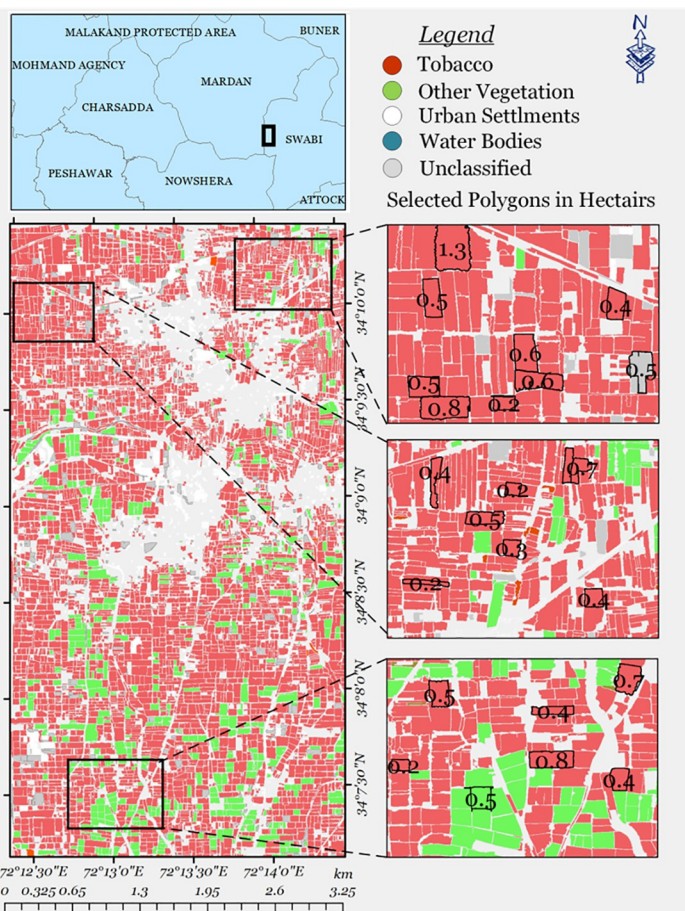

**Fig 15. Parcel processed classification map.**

entails data collection and curation by stakeholders, employment of open-source tools, like AROSICS for multi-sensor geometric correction and SAM for vegetation parcels segmentation, in synergy with Bi-LSTM for crop classification. Validation results coupled with post-classification validation surveys, have confirmed the field-level accuracy of the model-based results. While considering datasets for major crops of both Kharif (*DatasetA*) and Rabi (*DatasetB*), the Bi-LSTM model has yielded impressive results with validation accuracy of 96% for Kharif *DatasetA* and 94% for rabi *DatasetB*, demonstrating the effectiveness of the approach. To further address the challenge of smallholder farms, Pre-trained Vision Transformer (ViT) encoder and a lightweight transformer-based decoder for segmentation of land parcels is utilized. These parcels are then refined through the presence of a majority filter by overlaying them on classified map. Surveys to retrieve ground truth data were conducted, utilizing our custom developed "GeoSurvey" mobile application. This data has been meticulously split into training and testing datasets, ensuring a systematic approach to avoid spatial mixing. In methodology, multi-sensor data of Sentinel-2 and Planet-Scope were effectively fused to enhance the spatial resolution of former data 10m channels to latter's 3 meters resolution, a critical improvement for precise agricultural analysis. Geometric correction has been carried out between corresponding scenes using frequency domain approach of AROSICS. Experiments were conducted on Precision 7820 Tower, powered by Intel Xeon Silver 4208, 64 Gigabyte of

RAM and Nvidia Quadro RTX 8000 48GB. The results obtained underscore the potential for precision agriculture and in-formed policy making. Looking ahead, this work opens doors to further exploration and the application of similar methodologies to empower smallholder farmers and enhance agricultural productivity, ultimately contributing to the betterment of agrarian economies and the well-being of their populations. Further research is needed for utilization of opendata satellite systems, like Sentinel-1 Synthetic Aperture Radar, which will greatly enhance the effectiveness of the employed approach".

## Acknowledgments

This research is conducted within the framework of the National Center for Big Data and Cloud Computing (NCBC). We gratefully acknowledge the reception of a data grant from Planet-Scope as part of the 'Education and Research' initiative.

## Author Contributions

**Conceptualization:** Waleed Khan, Nasru Minallah, Atiq ur Rehman, Tareq Al-Ansari, Amine Bermak.

**Data curation:** Waleed Khan, Nasru Minallah, Madiha Sher, Mahmood Ali khan, Atiq ur Rehman.

**Formal analysis:** Waleed Khan, Nasru Minallah, Atiq ur Rehman, Amine Bermak.

**Funding acquisition:** Atiq ur Rehman.

**Investigation:** Waleed Khan, Madiha Sher, Atiq ur Rehman, Tareq Al-Ansari, Amine Bermak.

**Methodology:** Waleed Khan, Madiha Sher, Mahmood Ali khan, Amine Bermak.

**Resources:** Waleed Khan, Tareq Al-Ansari.

**Software:** Waleed Khan, Nasru Minallah, Madiha Sher, Mahmood Ali khan, Amine Bermak.

**Supervision:** Nasru Minallah, Atiq ur Rehman.

**Validation:** Waleed Khan, Nasru Minallah, Mahmood Ali khan, Tareq Al-Ansari.

**Visualization:** Waleed Khan, Mahmood Ali khan, Atiq ur Rehman.

**Writing – original draft:** Waleed Khan, Nasru Minallah, Madiha Sher.

**Writing – review & editing:** Nasru Minallah, Mahmood Ali khan, Atiq ur Rehman, Tareq Al-Ansari, Amine Bermak.

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
