## [Decision Letter · Decision Letter 0]

8 Jan 2024

PONE-D-23-40002Advancing Crop Classification in Smallholder Agriculture: A Multifaceted Approach Combining Frequency-Domain Image Co-Registration, Transformer-Based Parcel Segmentation, and Bi-LSTM for Crop ClassificationPLOS ONE

Dear Dr. Rehman,

Thank you for submitting your manuscript to PLOS ONE. After careful consideration, we feel that it has merit but does not fully meet PLOS ONE’s publication criteria as it currently stands. Therefore, we invite you to submit a revised version of the manuscript that addresses the points raised during the review process. Please make the changes i naccordance to the comments and suggestions by the reviewrs. When makin g changes please keep in mind that the readers should be able to repeat the research and obtain the same or similar results. Please submit your revised manuscript by Feb 22 2024 11:59PM. If you will need more time than this to complete your revisions, please reply to this message or contact the journal office at plosone@plos.org. Please include the following items when submitting your revised manuscript:A rebuttal letter that responds to each point raised by the academic editor and reviewer(s). You should upload this letter as a separate file labeled 'Response to Reviewers'.A marked-up copy of your manuscript that highlights changes made to the original version. You should upload this as a separate file labeled 'Revised Manuscript with Track Changes'.An unmarked version of your revised paper without tracked changes. You should upload this as a separate file labeled 'Manuscript'.If applicable, we recommend that you deposit your laboratory protocols in protocols.io to enhance the reproducibility of your results. Protocols.io assigns your protocol its own identifier (DOI) so that it can be cited independently in the future. For instructions see: https://journals.plos.org/plosone/s/submission-guidelines#loc-laboratory-protocols. Additionally, PLOS ONE offers an option for publishing peer-reviewed Lab Protocol articles, which describe protocols hosted on protocols.io. Read more information on sharing protocols at https://plos.org/protocols?utm_medium=editorial-email&utm_source=authorletters&utm_campaign=protocols.

We look forward to receiving your revised manuscript.

Kind regards,

Tomo Popovic, Ph.D.

Academic Editor

PLOS ONE

 [This research work is funded in-part by Qatar National Research Fund (QNRF) through grant no. MME01-0922-190049. The findings herein reflect the work and are solely the responsibility of the authors.].  

[This research work is funded in-part by Qatar National Research Fund (QNRF) through 

grant no. MME01-0922-190049. The findings herein reflect the work and are solely the 

responsibility of the authors]

 [This research work is funded in-part by Qatar National Research Fund (QNRF) through grant no. MME01-0922-190049. The findings herein reflect the work and are solely the responsibility of the authors.]. 

5. We note that Figure 4 and 8 in your submission contain copyrighted images. All PLOS content is published under the Creative Commons Attribution License (CC BY 4.0), which means that the manuscript, images, and Supporting Information files will be freely available online, and any third party is permitted to access, download, copy, distribute, and use these materials in any way, even commercially, with proper attribution. For more information, see our copyright guidelines: http://journals.plos.org/plosone/s/licenses-and-copyright.

A. You may seek permission from the original copyright holder of Figure 4 and 8 to publish the content specifically under the CC BY 4.0 license. 

6. We note that Figure 1, 2, 3, 8, 9, 10 and 13 in your submission contain [map/satellite] images which may be copyrighted. All PLOS content is published under the Creative Commons Attribution License (CC BY 4.0), which means that the manuscript, images, and Supporting Information files will be freely available online, and any third party is permitted to access, download, copy, distribute, and use these materials in any way, even commercially, with proper attribution. For these reasons, we cannot publish previously copyrighted maps or satellite images created using proprietary data, such as Google software (Google Maps, Street View, and Earth). For more information, see our copyright guidelines: http://journals.plos.org/plosone/s/licenses-and-copyright.

a. You may seek permission from the original copyright holder of Figure 1, 2, 3, 8, 9, 10 and 13 to publish the content specifically under the CC BY 4.0 license.  

Additional Editor Comments:

Please check the comments and suggestions rfom the reviewers.

In addition, when making changes to the paper keep in mind that the readers should be able to repeat the research and obtain the same or similar results.

Reviewers' comments:

Reviewer's Responses to Questions

**Comments to the Author**

1. Is the manuscript technically sound, and do the data support the conclusions?

Reviewer #1: Yes

Reviewer #2: Yes

2. Has the statistical analysis been performed appropriately and rigorously? 

Reviewer #1: Yes

Reviewer #2: N/A

3. Have the authors made all data underlying the findings in their manuscript fully available?

Reviewer #1: No

Reviewer #2: Yes

4. Is the manuscript presented in an intelligible fashion and written in standard English?

Reviewer #1: Yes

Reviewer #2: Yes

5. Review Comments to the Author

Reviewer #1: This research deals with the topic of Agricultural Remote Sensing in terms of overcoming obstacles in enhancing smallholder economies' agricultural monitoring to mitigate losses such as diminishing farm sizes, lack of reliable data sets, and high costs related to commercial satellite imagery using novel approaches.

Besides the previously recognized challenges for smallholder agriculture, including lack of reliable datasets, geometric error in multi-sensor satellite images, land fragmentation resulting in holdings diminishing in size, inadequate spatial resolution of open data satellites for the field sizes, clouds that can hinder the capturing of satellite images authors emphasize problems specific for country of Pakistan. Namely, the agricultural production of this country contributes 20-22% of its gross domestic product (GDP) to its economy. With a growing population of over 230 million, it is becoming difficult to sustain its own agricultural production, allowing imports of various agricultural commodities. Also, another problem caused by over-hoarding by stakeholders is the growth of counterfeit crops and unlicensed or unaccounted growth of cash crops like tobacco. The country's lack of state-of-the-art technological machinery and the persistence of outdated farming methods are major causes of low yields and financial struggles for smallholders, together making a pivotal point of the described experimentation.

The possible solution for addressing challenges associated with remote sensing-based crop mapping in smallholder agriculture farms includes open source tools and data used for inter and intra-sensor image registration, Segment Anything Model for Geospatial (SAM-GEOs) for delineated vegetation parceling, and a Bidirectional-Long Short-Term Memory-based (Bi-LSTM) deep learning model, developed and trained for crop classification. The proposed method achieves respectable results with accuracy of more than 94% and 96% for validation sets of two data sets collected in the field from the area covering 4 major districts of Pakistan, including a total area of 580000 hectares, during 2 growing seasons.

The manuscript describes a technically sound piece of scientific research with data that supports the conclusions. The experiments have been conducted rigorously, with appropriate controls, replication, and sample sizes. The manuscript presented in an intelligible fashion and written in standard English, but with some typographical errors, that should be corrected. However, manuscript has some minor shortcomings that should be removed. Although, the manuscript contains names of the satellites' from which images are used, the direct link to these images is missing. Also, authors should make link to a numerical data .csv file derived from satellite images. The author's motivation for choosing the Bi-LSTM method is not stated in the paper, i.e. ignoring other methods previously applied in this area. Also, the precision measures obtained in this research are slightly worse than in earlier researches that used the same method. The existing research could be expanded by determining the reasons for the stated deviation, because 2-3% precision could be high considering the share of agriculture in the total GDP.

Reviewer #2: Authors wrote interesting papers about Crop Classification in Smallholder Agriculture using deep learning techniques. Comments and questions which can be discussed in following versions are:

- Introduction is quite big, but interesting. Some parts can be moved to “related work” section

- You used Bi-LSTM DL model. Have you tried to run same model with same dataset using classic LSTM model? Can you compare results (precision and similar) and execution time?

- You mentioned different satellites. Can you compare their resolution/ratio on one place (especially newer versions (like Sentiel-1/2 …) and their dataset size?

- Model parameters are shown at “Model specification” section. Have you tried model with different parameters? Can you compare results (precision and similar) and execution time?

Development of mobile and web-based application for utilizing for collecting and curating these data and the presentation of the results is very praiseworthy.

6. PLOS authors have the option to publish the peer review history of their article (what does this mean?). If published, this will include your full peer review and any attached files.

Reviewer #1: **Yes: **Marko Grebović

Reviewer #2: No

---

## [Author Response · Author response to Decision Letter 0]

1 Feb 2024

The response is attached as a separate file because it contains figures.

---

## [Editor Report · Decision Letter 1]

9 Feb 2024

Advancing Crop Classification in Smallholder Agriculture: A Multifaceted Approach Combining Frequency-Domain Image Co-Registration, Transformer-Based Parcel Segmentation, and Bi-LSTM for Crop Classification

PONE-D-23-40002R1

Dear Dr. Rehman,

We’re pleased to inform you that your manuscript has been judged scientifically suitable for publication and will be formally accepted for publication once it meets all outstanding technical requirements.

Kind regards,

Tomo Popovic, Ph.D.

Academic Editor

PLOS ONE
---

## [Editor Report · Acceptance letter]

21 Feb 2024

PONE-D-23-40002R1 

PLOS ONE

Dear Dr. Rehman, 

I'm pleased to inform you that your manuscript has been deemed suitable for publication in PLOS ONE. Congratulations! Your manuscript is now being handed over to our production team.

Kind regards, 

on behalf of

Prof. Tomo Popovic 

Academic Editor

PLOS ONE